# Improving the Welfare of Companion Dogs—Is Owner Education the Solution?

**DOI:** 10.3390/ani9090662

**Published:** 2019-09-06

**Authors:** Izzie Philpotts, Justin Dillon, Nicola Rooney

**Affiliations:** 1Animal Behaviour and Welfare Group, Bristol Veterinary School, University of Bristol, Dolberry Building, Langford BS40 5DU, UK; 2Graduate School of Education, University of Exeter, St Luke’s Campus, Heavitree Road, Exeter EX1 2LU, UK

**Keywords:** dog, education, owner, ownership practices, welfare

## Abstract

**Simple Summary:**

The welfare of most dogs living in homes is largely unknown. However, national surveys carried out by animal welfare charities and findings by animal welfare researchers have shown significant deterioration in some key aspects of dog welfare. For example, more dogs presenting to vets with behavioural problems, obesity, and ill-health due to poor breeding practices. This means that some dogs are suffering due to their owners’ behaviours or ownership practices. Educating dog owners as to how best to look after their dogs is, and has been seen by many, as key to improving the welfare of dogs living in homes. However, the concept of education, the context in which it occurs, and the lack of systematic evaluation of the effectiveness of education interventions means that nobody really knows if this approach works. This paper explores these concepts and draws together a wide range of sources of information to highlight some of the complexities of improving dog welfare by educating owners.

**Abstract:**

Vets, animal welfare charities, and researchers have frequently cited educating owners as a necessity for improving the welfare of companion dogs. The assumption that improving an owner’s knowledge through an education intervention subsequently results in improvements in the welfare of the dog appears reasonable. However, the complexity of dog welfare and dog ownership and the context in which these relationships occur is rapidly changing. Psychology has demonstrated that humans are complex, with values, attitudes, and beliefs influencing our behaviours as much as knowledge and understanding. Equally, the context in which we individuals and our dogs live is rapidly changing and responding to evolving societal and cultural norms. Therefore, we seek to understand education’s effectiveness as an approach to improving welfare through exploring and understanding these complexities, in conjunction with the relevant research from the disciplines of science education and communication. We argue that well designed and rigorously evaluated education interventions can play a part in the challenge of improving welfare, but that these may have limited scope, and welfare scientists could further consider extending cross-disciplinary, cross-boundary working, and research in order to improve the welfare of companion dogs.

## 1. Introduction

This paper aims to explore the extent to which educating owners is an effective strategy for improving the welfare of companion dogs. Firstly, information relating to dog welfare, needs, and ownership practices will be reviewed. The concepts and components of education in the context of dog ownership will be examined and some relevant findings from science education and science communication literature will be presented. Finally, consideration will be given to the role that education can, and cannot play, in increasing dog owners’ knowledge, behaviours, and ownership practices to improve the welfare of their companion animals.

## 2. Background

Despite the introduction of compulsory microchipping in the United Kingdom (UK) in April 2016, it remains unknown how many dogs are living in homes. As such, the welfare state of each of these dogs is also unknown [1]. According to the Pet Food Manufacturers Association (PFMA) in 2018 there were 9 m dogs living in households in the UK [2]. As much as 26% of the UK population are thought to own dogs, with an average of 1.4 per household, and thus 6.6 m households with dogs [2]. The People’s Dispensary for Sick Animals (PDSA) Animal Wellbeing (PAW) annual report for 2018 reported that 24% of the UK’s adult population have a dog, which suggests that there may be up to 8.9 m dogs [3]. However, the Royal Society for the Prevention of Cruelty to Animals’ (RSPCA) #DogKind Report estimated that there are 11.5 m dogs [4]. When compared to the rest of the European Union (EU), the UK is ranked only 12th in the number of households owning at least one dog [5]. Romania and Poland (42%), Czech Republic (41%), Lithuania (37%), and Portugal (36%) make up the top five dog owning EU countries [5].

## 3. Welfare Science and Dog Welfare

Animal welfare is widely acknowledged as being complex multidisciplinary and multifactorial with scientific, biological, psychological, ethological, ethical, philosophical, social, cultural, economic, and political influences and consequences [6]. The impact of humans on all animals and their environment ensures that the relationship with people is central to animal welfare. It is also widely accepted that understanding and ensuring an individual domestic animal’s welfare needs are met throughout its lifecycle is a human responsibility [7,8].

Dogs have been associated with humans for many thousands of years [9]. They continue to fulfill many roles in different societies, but the majority are now classified as companion animals and share a unique relationship with people living in their homes [1].

Consideration of dog welfare involves not only ethical, moral, and philosophical debate and consideration, but must also result in practical and effective outcomes [10]. Welfare requires both a values and science-based approach [11]. Early approaches to defining welfare were largely focused on the exclusion of poor welfare or negative states. More recently, the concept of animal welfare has been viewed as a continuum between negative and positive welfare [12,13]. An animal’s experiences represent its welfare status, which can range from very poor to very good, as welfare is not something that can be done to someone or something, but is a state of being [14,15].

Traditionally there have been three distinct approaches to defining welfare [10]. The first approach is based on veterinary science and it focuses on biological and physiological functions affecting health [16]. These are all processes that can be objectively observed and measured. The second is a neuropsychological approach that considers the animal’s ‘feelings’ [17]. It attempts to understand the animal’s subjective experience of the world or its ‘affective state’. If an animal experiences a generally positive state, it is thought to experience good welfare. The third approach is based on the concept of natural living and considers wellbeing to be good if an animal can perform its natural behaviours, express its natural needs, inclinations, and ‘purpose’. The use of one approach to the exclusion of another can often result in inconsistent end points and conflicting information, as has been demonstrated for the dog [1]. Therefore, the most generally accepted conclusion is that a good approach to animal welfare considers all three approaches [17].

Despite reaching some consensus on an applied definition of welfare, ongoing academic debate continues. We continue to develop understanding and question the complex concept of what constitutes human health and welfare and the implications and impact of mental, as well as physical, health. Therefore, logic and, to an extent, anthropomorphism, suggest that these factors be considered and applied to debates in animal welfare science. Additionally, the political relevance of animal welfare science is strongly (if not exclusively) based on societal concerns regarding how animals are treated [12]. Therefore, increased levels of companion animal ownership and a media-driven increased interest in animal welfare more generally, have continued to drive popular interest in promoting the concept of animals being ‘happy’ [18,19]. Equally, our limited, but increasing, understanding of animal behaviour, cognition, and emotional states would suggest that as that body of knowledge continues to develop, we should be continually adapting our definitions of what is considered ‘acceptable’ or ‘good’ animal welfare.

Animal welfare is seen as a ‘new science’ by many and the development of companion animal welfare science even newer [20]. Due to the complexities of the interactions between humans and other animals, it has also be viewed as a somewhat disordered science, with few clear disciplinary boundaries [21,22]. Companion animal welfare and, most notably, dog welfare, has received increasing amounts of attention from many sources in recent years [23,24,25]. Research has often previously focused on how animals, including dogs, help or benefit humans, more than how well owners manage to provide and care for their dogs [26,27,28]. However, despite increases in research, media attention, and philosophical debate, there continues to remain a lack of consensus as to what constitutes good welfare for dogs [25].

There are also some who argue that, due to the intensive selective breeding of dogs over a significant number of years, dogs are now so far removed from their wolf ancestors that we cannot presume to consider their welfare in the same way as other animals [29]. Equally, through ongoing research and improving technological developments, our understanding of dogs’ abilities and their unique relationship with humans still remains in its infancy [30]. The co-evolution of both species and the significant potential for anthropomorphism and anthropocentrism makes defining, measuring, and researching welfare and quality of life in this animal group even more complex [31]. This complexity underpins the arguments that we advance in this paper.

## 4. What Are Dogs’ Needs and Do Owners Meet Them?

In England and Wales, Section 9 of the Animal Welfare Act [32] states that those responsible for any animal must take reasonable steps to ensure that they meet its needs. This includes the need: (a) for a suitable environment; (b) for a suitable diet; (c) to exhibit normal behaviour patterns; (d) to be housed with or apart from other animals; and, (e) to be protected from pain, suffering, injury, and disease. These five welfare needs are reflected in policy in many countries. In England and Wales, they are enforceable by law and alongside additional clauses and aim to protect animals from abuse, cruelty, poor treatment, or neglect by people. Significant volumes of guidance have been produced by various stakeholders for dog owners, and those who work with animal charities, local councils, and the law to advise owners and carers how best to meet dogs’ needs [33,34,35]. Despite the extensive availability of information, some of which is based on research evidence, some of which is not, it appears many owners are often unaware or (largely) choose to disregard their obligations to their pet dog [36].

Interestingly, the majority of primary (elementary) schools (60%) and secondary (high) schools (85%) do not teach about caring for pets, according to research that was commissioned by the PFMA in 2012 [37]. However, the RSPCA continues to campaign for education about caring for pets to be made part of the National Curriculum in England and Wales arguing that it would increase the proportion of the public aware of legislation such as the Animal Welfare Act 2006 and subsequently improve the welfare of pets [38].

The regulation of dog ownership is often considered as one approach for monitoring dog populations and improving the levels of welfare. Dog ownership licensing was abolished in the UK in 1987, but there have repeatedly been calls for its reimplementation. Other countries have varying levels of regulation. Switzerland is thought to have one of the most regulated ownership requirements having introduced an ownership ‘test’ in 2008. Equally, Northern Ireland, Canada, Sweden, and Germany, among others, have more regulation over dogs and owners. This compares to other countries where there is very little regulation and large free-roaming dog populations [39]. The effectiveness of these different approaches to regulating (or not) ownership does not appear to have been evaluated. As such, it remains unclear as to whether increasing ownership regulation, licensing, or taxing improves ownership practices or dog welfare. 

Meeting the welfare needs of dogs is often closely linked to ‘responsible dog ownership’. In the UK, this concept has been promoted by the Kennel Club [40], the British Veterinary Association [41], and other animal charities and councils. Responsible ownership campaigns have encouraged owners to meet their legal requirements not just meeting the five welfare needs but also engaging in control measures. These measures may include keeping dogs on lead where requested, picking up faeces, and promoting the importance of training. These campaigns have attempted to reinforce the responsibility and commitment of ownership, but they may have blurred the concepts of good welfare with responsible ownership practices. Nonetheless, the success or impact of these campaigns does not appear to have been evaluated and, as such, their effectiveness remains unknown. Interestingly, a large-scale Australian survey by Rohlf, et al. [42] showed that even committed dog owners failed to comply with some responsible dog ownership practices. They discovered a relationship between owners’ attitudes and behaviours and suggested that normative beliefs and peer pressures may play a role in forming and changing behaviours.

Dog ownership throughout much of the world is now a lifestyle choice. Given that up to 26% of homes in the UK contain dogs [2], dog owners or people living with dogs can potentially come from all walks of life. Researchers, such as Bennett and Rohlf [43], Diesel, et al. [44], Garrison and Weiss [45], among others, have attempted to identify correlations or patterns arising from dog owners’ social determinants, such as gender, age, social class, levels of education and income, among others, and their ownership practices. Others (e.g., Brown and McLean [46], Cimarelli, et al. [47], and Cloutier and Peetz [48]), have attempted to associate owner personality characteristics, interaction styles, or attitudes and belief systems associated with different levels of animal welfare. Probably unsurprisingly, the findings from these studies have varied considerably; and, whilst correlations may show a relationship between demographic factors and indicators of welfare, they cannot determine any causal effect, so any conclusions drawn need interpreting with caution.

As it is owner behaviour that is likely to have most impact on a dog, determining ownership practices is likely to be valuable in developing an understanding of the welfare of dogs living in homes. Large-scale UK [4,49] and Australian [50,51,52] surveys have all reported significant variations in ownership practices and behaviours. Researchers have repeatedly implied that the welfare needs of many dogs living in homes are not being consistently met, based on those variations.

Investigation into the welfare of domestic dogs in the UK mainly takes place though large-scale surveys commissioned by animal welfare charities. The PDSA’s PAW Reports, conducted in collaboration with YouGov, have provided an annual national survey of pet owners and veterinarians since 2011. These reports consistently show a shortfall in knowledge among many owners as well as providing evidence of changes in knowledge, understanding, and ownership practice over time. For example, 69% of pet owners frequently underestimate the cost of owning a dog, with research by the PDSA estimating that a dog can cost between £70–105 per month depending on its size [3]. This is a significant cost, as the average UK household spends just over £3000 per year or £250 per month on groceries, according to the Office for National Statistics [53]. The reports also frequently show a mismatch between owners’ knowledge and views and their actions. For example, dog owners acknowledged the importance of training and socialising dogs, yet very few actually undertook any training with their dogs during the first six months of their lives [49]. Similar mismatches in dog owner knowledge and behaviours have also been highlighted in the RSPCA’s #DogKind Report [4]. 

Many vets, who were surveyed as part of the annual PAW Reports, describe owners’ lack of knowledge and understanding as a major welfare limitation for dogs. However, these surveys findings are often limited by sampling methods, question and response design, inaccurate reporting of information from owners, and owner’s interpretations of their dog’s behaviours and experiences. Individual owners’ values, attitudes, and beliefs around animals and their place in the home, vary significantly [54] and will impact on ownership practices and responses. A minority of owners may intentionally cause harm or neglect. Others can cause unintentional harm or neglect, often as a result of misinterpretation of their dog’s behaviours or needs, or by relying on poor quality information or advice [55] or previous ownership experiences. Research shows that owners (and some experts) have difficulty interpreting their dog’s behaviours [56,57,58], and therefore may not understand or accurately interpret the causes and implications of some of their behaviours. As behaviours that dogs display can provide indicators to their welfare state, many owners may not see that their dog may be suffering. Whilst most owners are likely to show empathy and compassion for their dog, this may be misdirected by anthropomorphism or through the misinterpretation of those behaviours [31]. Equally saying or feeling something does not necessarily result in doing. For example, some owners may be aware that their dog has a fear of fireworks and loud noises, but do not seek to improve the problem for their dog [4].

When collecting information on dogs’ behaviours through online surveys, it may be difficult for researchers to establish the differences between reported behaviours and actual behaviours, as well as ‘why’ or ‘why not’ behaviours occur. One-off survey findings can only provide information that is based on a snapshot in time; the needs of both dog and owner can change significantly over a dog’s life-cycle. 

Despite the limitations of the scope of these surveys, they continue to provide the best current indication of the number of and welfare state of companion dogs. They also raise concerns regarding the ongoing welfare of some dogs living in homes. Survey findings have also been corroborated by research findings and reports from dog welfare experts. These suggest that some aspects of dog welfare have deteriorated rather than improved in recent years. The five most commonly reported dog welfare concerns relating to ownership practices are:pedigree or poor dog breeding practices [59,60,61,62];obesity [63,64,65,66];dog behaviour and training [67,68,69,70];dog purchasing and relinquishing behaviours [49,62,71,72]; and,dog companionship or being left alone for extended periods of time [4,49,73].

The extensively reported deterioration in these five elements of dog welfare appears to have continued, despite the substantial amounts of information available to owners, the significant funds spent on campaigns by dog and animal welfare charities, and vets becoming more aware and engaged with the welfare issues that dogs are facing. Dog ownership norms and expectations are now very different to those of dog owners 50, 100, and 1000 years ago. These changes are likely to be significantly affected by the contexts in which those individual relationships occur. McGreevy and Bennett [74] discuss the ‘challenges and paradoxes in the companion animal niche’, highlighting the conflict between what humans currently demand from companion animals and our abilities to meet the animal’s needs. For example, we are selectively and extensively breeding genetically and anatomically unhealthy dogs, many of whom suffer throughout their lives [59,75]. Some owners spend significant amounts of money on dog clothing and grooming under the impression that it makes their dogs ‘happy’ [74].

A significant number of dogs are now obese and suffer significant health problems and reduced quality of life as a result of their weight [63,66,76]. Owners, trainers, and behaviourists continue to work with dominance-based theories in their practice, despite current evidence of the detrimental welfare consequences [69,70,77]. Owners also struggle to interpret their dog’s behaviours [56,57,78] and they often fail to understand serious signs or symptoms of common diseases in older dogs [79]. Many owners go out to work leaving their dogs without company for significant amounts of time [73,80]. However, most of these dog owners would probably claim that they ‘love their dogs’ and that they would ‘do anything for them’ and that they lead ‘happy’ lives. 

There is no one ‘perfect’ way to care for all dogs, because every dog, and every situation, is different [81]. All dogs and owners are individuals whose needs change over time. Owning and providing good welfare for some dogs can be very challenging and complex. In most cases, owners do not deliberately set out to cause harm or suffering, but this can occur due to a lack of understanding of their animal’s needs, the importance of disease prevention, and the responsibilities of dog ownership [36]. These conclusions then result in the call for owners to be ‘educated’ about things that they do not appear to ‘understand’ [82]. Equally, ‘education’ has long been stated as priority for all major animal and dog welfare charities. Many charities spend significant amounts on campaigns that aim to educate the public about certain elements of dog ownership and animal welfare in general. However, the rationale or evidence-base for the approaches used is not often clear and the effectiveness rarely effectively evaluated. Accordingly, we ask, to what extent is education the answer?

## 5. How Effective Is Educating Dog Owners?

The impact of owners on their dog’s welfare is undeniable. The assumption that to improve the welfare of dogs living in homes, welfare experts, or advocates need to educate owners or the dog owning public makes sense when taken at face value. However, the term education within the context of improving companion animal welfare often appears to be asserted as a ‘fix all’ approach by those with significant animal welfare expertise. However, this may be due to a limited understanding of what education is and does, and can or cannot achieve. Exploring the role of education within the context of improving dog ownership practices requires the integration of multiple perspectives and a variety of disciplinary approaches and sources of evidence. In the next part of the paper, we aim to integrate and define the complexity of those multiple perspectives and describe and evaluate sources of evidence and current approaches to education. 

Society and the media often portray dogs as being integral to the family [83,84]. In the UK, buying a dog is relatively easy; you can go online, find one, view it, and buy it instantly. You do not need a licence, often you do not need a home check, you just need to decide you want one and be willing to pay. It is almost just as easy to dispose of the dog to a charity that will continue to care for it when you are no longer able or wish to keep it. The perception appears to be that owning a dog is ‘normal’ and ‘everybody does it’ therefore it must be easy. If owning a dog is easy, then there is the presumption or assumption that anyone can look after any dog. If individuals lived with dogs whilst growing up, then they surely know how to care for all dogs. Equally, if individuals have owned dogs in the past, there is the assumption that they too know how to care for all dogs. Hence, do most dog owners think that they need ‘educating’ as to how to look after their dog? The evidence suggests otherwise. The majority of owners think that because they ‘love their dog’ and ‘would do anything for them’ this makes them good owners who must therefore have ‘happy’ dogs. 

However, we have established that dog experts or advocates, such as vets, animal welfare scientists, and national surveys commissioned by animal welfare charities have all repeatedly described the welfare of many dogs living in homes as inadequate or as could be improved. Therefore, there is a potential conflict of opinion as to the need for education in the first instance. How do you educate someone who does not think they need it? Or does not see the problem? Or can see the problem, but does not think it relates to them or their dog? Even then, if owners know what they should be doing, that knowledge does not necessarily result in the preferred behaviour and result in good ownership practices and improved welfare for the individual dog. For example, the RSPCA Being #DogKind Report [4] suggested that 87% respondents agreed or strongly agreed that dogs need and value the company of people; however, 22% of dogs were reported to spend more than four hours alone, the maximum amount of time charities deem acceptable [85], during the average weekday. Even when education does result in increased knowledge, it does not necessarily result in changes to behaviour.

## 6. What Is Education?

Education is defined in the as “the process of receiving or giving systematic instruction, especially at a school or university” [86]. 

The process of education, as exemplified by current practice in the area of dog ownership, can be simplified into three components (Figure 1), each of which will now be explored. 

Learning is frequently and largely linked with the concept of education and teaching, but it is also linked to an individual’s experiences outside of the formal learning environment. Learning is defined as “the acquisition of knowledge or skills through study, experience, or being taught” (ibid.). Additionally, teaching is defined as “Ideas or principles taught by an authority” (ibid.). An individual’s learning, alongside teaching methods (pedagogy) has been extensively investigated and defined by educational psychologists [87]. However, these definitions and theories all relate to learning and teaching within recognised educational contexts, such as schools, colleges, and universities (ibid.). There is little that relates these definitions to the general public in alternative, non-traditional, non-formal learning environments or contexts. There are no regulated schools or colleges for potential dog owners and no requirement for owners to learn about or understand their dogs. However, in the UK, there are ‘dog schools’, such as the ‘Dogs Trust Dog Schools’ [88], which may provide valuable support and education for dog owners (and their dogs), but these are often limited by factors, such as owner’s ability or motivation to attend, cost, class capacity, and geographical availability. Equally, their value is yet unmeasured. Some ‘schools’ provide classes that can be of varying quality and value depending on the skills of the trainer and the training approaches taken.

There is very little regulation, beyond reported law breaking, over the behaviours that occur in people’s homes. There is an expectation by law that all dogs in the UK are microchipped [89], but this process is virtually impossible to enforce in isolation. It is good practice that dogs are registered with vets, but again this is impossible to enforce or regulate. Therefore, the structure or regulation that is suggested in the dictionary definition of education does not currently exist in the context of dog owner education. Accordingly, who, what, or where are the current dog owner educators, teachers, instructors, authorities, communicators, or influencers? We proposed that these are multiple (Figure 2), and each of these will now be explored.

### 6.1. Dog Professionals and Perceived Dog Experts

Qualified dog professionals include vets, vet nurses, physiotherapists, some animal carers, dog trainers, behaviourist, groomers, hydro-therapists, and complementary therapists. In the UK, vets and vet nurses are subject to formal, regulated training programmes and are bound by rules of professional conduct [90]. Traditionally professional training programmes have focused more on an animal’s physical health (medical model), but more recently their training and roles have considerably evolved to become more holistic and to consider animals’ welfare and behaviour [13]. However, research shows that vets in the UK frequently feel that they could and/or should do more about key dog welfare issues [15]. Roshier and McBride [55] conducted a small-scale analysis of veterinary consultations during annual boosters and showed that of 58 problem behaviours identified by dog owners, only 10 were discussed and none was fully explored by the veterinarian. They concluded that the vets required further support and training in dealing with behavioural problems. Similarly, Belshaw, Robinson, Dean, and Brennan [91] interviewed 14 vets and 15 dog and cat owners following preventative healthcare consultations. In a series of papers, they report that, even in these routine appointments, there was potential opportunity for vets to explore owner attitudes and inform or educate owners about good practice [91]. However, vets often felt under pressure due to a lack of time [92], and there was a mismatch in expectations with both parties often unable to achieve what they had hoped from these consultations [93].

Other researchers, such as Bard, et al. [94], have explored the effects of vet communication style on client behaviour change relating to farm animal welfare. They suggested that communication styles during consultations could make a difference to the effectiveness of advice or education on good animal welfare practice. These studies start to highlight some of the potential limitations that vets may have in their role as dog owner educators.

For most other perceived dog experts, there is very little regulation or training required. There are some extensively trained, highly experienced experts utilising evidence-based practice in their work. However, there are also many out there with no training and little experience that market themselves as experts in their field. There is no guarantee that the behaviourist or trainer the average dog owner finds online is in anyway a recognised expert. UK based accreditation schemes, such as the Association for the Study of Animal Behaviour [95] and Association of Pet Dog Trainers [96], attempt to ensure standards in behaviourists and trainers, respectively. However, a lack of awareness by the public and lack of regulation suggests that their impact has been limited. Equally, breeders found online are often considered to be experts in their breeding practices by puppy purchasers, having no guarantees. The impact of resources such as the ‘Puppy Contract’ [97] is not yet clear. Dog walkers and dog day-care providers may have little or no knowledge or experience of dogs whatsoever, yet many owners happily hand over their dogs to these individuals on a daily basis. Therefore, any advice or ‘education’ for owners by these unregulated, yet perceived to be experts, is likely to be somewhat hit or miss at best and could even result in reducing, rather than improving, the dog’s welfare.

### 6.2. Laws, Rules and Enforcers

Whilst enforcing the rules and the law, enforcers have a significant opportunity to educate dog owners regarding their ownership practices. The Animal Welfare Act [32] provides a legal framework to protect the welfare of animals in the UK. However, the annual PDSA PAW reports demonstrate that a significant number of dog (and other pet) owners are not aware of the Act or the five welfare needs that are defined within it [36,49], despite agreeing that all of the owners should be aware of them. The UKs Department for Environment, Farming, and Rural Affairs (DEFRA) provides rules and guidance on the movement of dogs and other animals and enforces the relevant laws. Local councils set rules and guidelines relating to where dogs can and cannot be walked, they provide bins for dog owners to dispose of waste, and wardens collect stray or lost dogs. RSPCA inspectors not only respond to concerns over animal welfare and, alongside the police, enforce the law relating to the Animal Welfare Act, but also rescue animals, inspect businesses involving animals, and respond to complaints of cruelty or neglect. However, the ‘average’ dog owner might never have cause to meet or interact with any of these laws, rules, or enforcers beyond seeing signs and dog waste bins in their local parks. Accordingly, whilst, for the small majority of dog owners, these laws, rules, and enforcers can act as an opportunity for education (e.g., via RSPCA improvement notices), the majority may never be in situations that would require them to engage in these education opportunities. 

### 6.3. Charities

Animal and dog welfare charities aim to play a significant role in educating the public about ownership. National charities, such as the RSPCA, Dogs Trust, PDSA, and the Kennel Club, are probably some of the highest profile, widely recognised brands of dog advocates and experts. This may largely be due to their media campaigns and marketing profiles, many of which are largely focused on providing education to dog owners and potential owners. Charities frequently report education as a main part of their charitable role in improving animal welfare and regularly launch high profile national campaigns. However, they rarely consider the meaning or effectiveness of this approach or evaluate the successes of their endeavours. When consideration is given to success, measures are often stated in terms of interactions with numbers of school children in a year, for example. However, the effectiveness or outcome of those interactions is less frequently quantified. Equally, campaigns, such as that launched in 2007, highlighting the welfare implications of pedigree dog breeding, appear to have little impact as trends in procuring brachycephalic dogs remain high [98]. 

### 6.4. Individuals

People learn from the people that they live with, grow up with, socialise with, work with, chat with on social media, etc. Their social circle informs their values, attitudes, and beliefs, which in turn impacts their behaviours. Rohlf, Bennett, Toukhsati, and Coleman [51] highlighted the influence of normative beliefs and expectations on ‘committed dog owners’ and suggested that peer influence was a significant indicator and dictator of their behaviours. They suggested that practices, such as microchipping, neutering, and socialisation, were more likely to occur if the views of friends and family supported these activities [51]. Equally, Kogan, et al. [99] investigated the use of the internet in finding pet health information and found that not just vets, but other pet owners, were rated as the most trustworthy sources of information by owners. 

### 6.5. Universities, Colleges and Training Course Providers

Animal welfare scientists, researchers, and academics all strive to enhance our understanding of, and improve, the welfare of animals. There are significant amounts of research being undertaken throughout the world with new and interesting knowledge being continually discovered. However, Ohl and van der Staay [12] suggested that significant animal suffering continues despite this acquisition of knowledge and that new knowledge may not reach those who need to know it. In the case of dog welfare, there are an increasing number of academic courses and training programmes for those that are interested in all elements of animal welfare. From a day’s training aimed at dog owners, to degrees and master’s programmes, veterinary and veterinary nurse training, behaviourists, physiotherapists, massage courses, first-aid courses are all now widely available. Regulation over the quality and accuracy of the content of some of these courses significantly varies depending upon the education providers.

### 6.6. Businesses

Businesses can significantly influence dog owners largely through advertising and marketing. Pet shops can influence market demands, dog food manufacturers can promote their brands, and dog trainers or dog walkers can promote their services through social media. As well as promoting their products or services, they are also educating, advising, suggesting, and normalising behaviours that are associated with their products or services. For example, dog food manufacturers produce advice for dog owners on food packaging as to how much to feed their dogs. In light of the current, significant dog obesity problem [63,64,100], it may be worth questioning how effective this information and approach is in providing the information owners need to feed their animals appropriately [100].

### 6.7. Media

The use of media in people’s lives has significantly increased with the rise in the use of the internet. The use of formal, structured, and regulated sharing of information can be of exceptional value, but the sharing of inaccurate or harmful information is of concern. Additionally, of significant potential interest is the unintended impact of some aspects of the media. For example, the role that advertising, films, TV, fashion, and celebrities play in influencing the purchasing behaviours of potential dog owners that has been repeatedly demonstrated [84,101,102].

A few years ago, a series of celebrities who were pictured with French Bull Dogs and Pugs likely contributed to a significant increase in the UK public purchasing these breeds. This trend was not due to an extensive advertising campaign by these breed groups, but simply the impact of seeing individuals in the media with these dogs. The potential for Brachycephalic Obstructive Airway Syndrome (BOAS) within these breeds [61] and the desire by owners for dogs to have a desired appearance has led to significant welfare problems for a large number of dogs that live in homes. 

In contrast, a study by Browne, et al. [103] examined the five best-selling dog training books in 2009 and 2012. They found significant differences across all five books with inconsistencies in the depth of information presented and vastly different training methods proposed. They suggested that the books reviewed did not perform as well as instruction manuals. The persistent promotion in the use of negative training methods known to be detrimental to welfare [69] and the popularity of these books is likely to contribute to poor dog training and ownership practices. Accordingly, even when owners attempt to educate themselves on the best way to train their dog, the information in the best-selling books might do more harm than good. However, somewhat surprisingly, the use of the internet by UK pet owners for finding information on animal healthcare has only recently started to be explored. Kogan, et al.’s [99] survey of pet owners reported that, of a sample of 571 respondents, 78.6% used the internet to find health information about their pets, but very few veterinarians actually recommended any websites to clients. This means that owners may well be looking at inaccurate information online and by veterinarians acknowledging that owners look for information online, they may be able to direct this searching to ensure that owners find the right information for their dog’s needs.

## 7. Educating the Owner—Issues, Contexts, and Challenges

The context in which dog owners may find or be exposed to opportunities for education varies considerably. It can often be through more formal methods or approaches (Figure 1), such as veterinary consultations or academic training programmes, but equally and likely far more frequently, through informal or indirect means, such as TV programmes, internet chat forums, or online videos. Equally, the physical environment in which dog owner education occurs varies from clinics, dog walks, pet shops, and dog training classes. However, it appears that many of these education opportunities or experiences occur within the home, largely from media and marketing sources, and can be entirely unregulated and inaccurate, potentially playing a role in reducing the welfare of dogs living in homes (Figure 2). 

Communication is defined as “The imparting or exchanging of information by speaking, writing, or using some other medium” [86]. In its simplest form, it is the process of transferring information from one place to another. Communication is fundamental to education and education cannot happen without communication. Information (or knowledge) transfer is a key part of the communication. The manner in which the information is communicated, as well as the information itself, is of relevance to the success or failure of the communication. Information must be easily available to the intended consumer. However, there is now access to vast quantities of information and to filter what is the ‘best’ or ‘most accurate’ or ‘most relevant’ information from an online search is almost impossible [79]. To make information accessible, it must be presented in the most appropriate format, at the right level and in the right language for the intended audience. Popular formats that information can present, includes words and/or pictures (printed or electronic), audio, or video, among others. The accuracy of information is of particular significance. Inaccuracies in information may be intentionally, or more often, unintentionally misleading. Unintentional inaccuracies in information might be due to a lack of understanding or misinterpretation of the subject, its complexities, and the evidence base. An example of this phenomenon still remains evident in dog training, with the concept of the alpha dog and negative forms of punishment still being used and promoted by self-appointed dog experts.

The intended purpose of the information being delivered is also worthy of consideration. Education and materials intended to educate are frequently linked with promotion or the sale of brands or products. For example, charities in the UK provide some high-quality educational materials for dog owners and potential owners, but each charity provides their own version from their own websites or stores or rehoming centres and often limits their information sources to their own branded products. How does an owner decide whose information to use given the volume of information available? Each individual will search for information from a source that they find to be credible and trustworthy [104]. Sources of inaccurate information from perceived to be trustworthy sources, are therefore a huge potential risk to those seeking accurate answers. 

Similar practice occurs with businesses, such as dog food manufacturers or retailers of pet products. Ultimately, who or where someone chooses to obtain their information from will depend on the effectiveness of the marketing strategy, making the brands or sources of information appear to be trustworthy, more than the quality of the information. Even then, no matter the quality of the information or how well (or not) it is communicated, the impact of the information will still depend on the individual receiving it. The individual still has to access, consume, understand, critique, interpret, and then accurately apply that information to his or her own individual circumstances. 

With many potential education initiatives, monitoring, or measuring the effectiveness can be challenging depending upon the approach taken, especially when there is limited understanding of cause and extent of the problem(s). There are likely to be many different variables that affect interventions, including changes in social norms, habits, and tastes, all of which may result in changes to attitudes and affect lifestyle choices. Education often aims to prevent behaviours from happening; the absence of something is very difficult to measure. Researchers have evaluated the effectiveness of some interventions on small research populations. One example being Blackwell, et al. [105], who provided written information and advice relating to separation related behaviours in dogs adopted from rehoming centres. The difference between those conducting research and more real-world education interventions is that researchers factor the need for clear aims, interventions, and outcome measures. Those with the motivation to educate the general public might be more focused on their desire to educate than to monitor or evaluate the effectiveness of process itself. Equally, there is no standardised assessment tool, theory, or model in which to work and no one-size intervention or evaluation method is likely to fit all of the approaches.

## 8. What About Dog Owners?

Understanding people has long been the pursuit of social and natural scientists. Psychologists, sociologists, neuroscientists, healthcare scientists, behaviourists, economists, advertisers, educationalists, researchers, and biologists, among many others, have attempted to understand how and why people think and do what they do. However, people are complicated, they can change over time and with experience, the context in which they function (society and culture), and their perceptions of the world also changes. Some of the factors that will influence individuals and therefore dog owners engaging with potential education opportunities include an individual’s age, gender, race, religion, social, and cultural identity. Other factors include previous experiences, habits, behaviours, emotions, perceptions, biases, self-efficacy, motivations and trust as well as values, attitudes, beliefs, personality, intelligence, cognition, self-awareness, physical ability, literacy, and analytical skills, etc.

Exploring the intricacies of what individuals think and do is very complex. Dog owners and people living with dogs represent approximately one-quarter of the population in the UK and significant additional numbers of individuals have owned or will own dogs. Subsequently, there are those who regularly interact with dogs, which makes dog ownership a community if not of general public relevance. However, it is not necessarily just about people and what they know or do not know, it is also about what they do or do not do that impacts on their dog’s welfare. People ought to know that their dog has needs in the first place, they need to consider those needs throughout its lifetime, showing empathy and compassion, without anthropomorphism, for their dog. However, dogs’ needs are complex, they vary considerably between individual dogs and over time. Owners need to understand and interpret their dog’s behaviours to be able to adapt their management strategies to its needs. Trying to understand what motivates people is complex in itself, but the interaction of two different species, often with the interaction of more than one ‘dog owner’ and, frequently, with more than one dog per household functioning in a changing society is a complex situation with multiple, changeable variables. Accordingly, what does the research say regarding whether education can make a difference in improving the welfare of dogs living in homes?

## 9. What Can We Learn from Animal Welfare Science, Science Education and Science Communication?

Academic questions regarding how animal welfare science relates to other science disciplines have long been debated [20,21]. Lund, et al. [22] provided a persuasive argument that animal welfare science, as a discipline, sits between the natural and social sciences. They suggested that companion animal welfare is an area well suited to cross-boundary working and could integrate research from and between behavioural, psychological, medical, sociological, economic, and ethical disciplines [22]. They also stated that whilst natural scientists are clearly central in achieving improvements in animal welfare, social scientists might be able to aid significantly in the understanding of the role of human behaviour and animals’ roles in society, as well as implementing solutions for improving animal welfare in practice [22]. When Lund, et al. [22] published their work, they highlighted the limited amount of cross-boundary working and despite some isolated examples of true interdisciplinary practice this appears to largely remain the case to date. Interestingly, Ohl and van der Staay [12] highlighted the lack of improvement to animal welfare and the limited consistent interdisciplinary or cross-disciplinary working may be one of the reasons for this. 

### 9.1. Science Education or Science Communication?

As animal welfare science is a scientific discipline, exploring the findings from the science education literature in the context of dog owner education provides one potentially significant opportunity for some cross-disciplinary working and integration. The discipline and concept of science education evolved in the 1950′s [106]. The priority of science education was educating the next generation(s) of scientists within education institutions, aiming to teach science content as well as improve scientific literacy, which therefore enables those individuals to respond critically to scientific debate and make informed decisions. In parallel, a separate discipline of science communication has since evolved. Science communication aims to educate the public through journalism, entertainment, and media, with the aim to engage the public with science. The underlying assumption often being that the more knowledge the public has about science, the more sympathy it has with science, the more likely the public’s decisions will support or agree with the scientific consensus. However, this assumption has since been shown to not necessarily be true [106]. Consequently, science communication, rather than science education as a discipline, may provide as much, if not more, useful sources of information when considering the concept of educating dog owners.

Historically science communication brings together branches of educational psychology, learning sciences, and science education. It also draws on social psychology, science, and technology to bring together the communication sciences. Science communication or engagement with science outside of the context of formal education initially focused on the public understanding of science (PUS), but has since evolved to the pursuit of public engagement with science (PES). Initially, there was an underlying presumption that when the public’s attitudes diverged from the scientifically accepted and well-established facts that this was due to a lack of information and limited scientific literacy; the logic was then that the public needed to be educated. This view has since been criticised by Nisbet and Scheufele [107] and it has been largely discredited by data showing only weak relationships between knowledge of science and attitudes to science [108]. Therefore, research and debate regarding the most effective approach to science communication remains ongoing.

### 9.2. Understanding Welfare Science

Science as a discipline is vast and limitless in depth and breadth, with nearly all concepts or specialties linking with others. However, specialities or disciplines have become narrowed and specialised to make these fields of knowledge manageable to the experts. As previously highlighted, this narrowing and restraining of specialities is clearly apparent within the animal welfare science field, as well as disciplines such as medicine and physics. Bromme and Goldman [104] debated the public’s ‘bounded understanding of science’. They explored the importance of researching how people make decisions in science domains, such as health, medicine, and the environment without having a deep and extensive understanding of the subjects. 

Bromme and Goldman’s [104] debate can be clearly extrapolated to dog owners making decisions that impact on dog welfare, such as neutering, vaccinating, feeding, or seeking appropriate help for training or behavioural problems. Their paper highlighted the challenges to understanding science, including the relevance of the information to the individual, the tentativeness of truth or ambiguity in many specific situations, distinguishing between non-science and science, and determining whether claims are true or false [104]. They described people’s limitations in knowledge and understanding as ‘bounded’. Bromme and Goldman suggested that we must acknowledge and understand that the way the public uses science information to make personal, professional, and civic decisions are rational responses to the inevitable limits of their understanding of science in order to improve education strategies. However, many scientific issues do require a far deeper understanding of scientific phenomena. Bromme and Goldman [104] suggested that in the public’s attempt to understand science, they tend to involve some scientific concepts, but then ignore many others that underlie the conceptual complexity. For example, owners attempt to train their dog to perform certain tricks through a learnt process, but they often fail to understand the way in which dogs learn best and the potential consequences of their method of training on their dog.

Kahan [109] stated that ‘never have human societies known so much… but agreed on so little’. He highlighted the paradox of the impact of science communication and suggested two not mutually exclusive theses, the public irrationality thesis and the cultural cognition thesis. The public irrationality thesis focused on the lack of scientific literacy by members of the public and their appraisal of information intuitively through fast-tracking unconscious emotions (heuristic thinking). He also suggested the public tends to overestimate dramatic sensational risks, such as terrorism and discount more remote, but consequential, risks, such as climate change. In terms of dog ownership practices, owners might overestimate the risk to dog welfare of severe abuse or neglect, resulting in near starvation of dogs, such as that regularly shown in the RSPCA’s ‘The Dog Rescuers’ television series [110], but concurrently discount the impact of their own dog’s obesity status to its welfare.

Kahan’s [109] cultural cognition thesis suggested that people tend to conform their assessments of evidence to various goals, which might not be accurate. These goals may include the need to reflect social identity, matching cultural peers, or common moral values, essentially agreeing with what they think everyone else is thinking. This view is supported by the findings of Rohlf, Bennett, Toukhsati, and Coleman’s [51] investigation into so-called committed dog owners in Australia. Their conclusions suggested that normative expectations played a significant element in their dog ownership behaviours. 

Similarly, Bromme and Goldman [104] proposed that people’s interests are not driven by the science that they think they understand. Accordingly, for owners who think that they know about dogs and dog ownership practices, they may be potentially less interested or engaged in media stories or campaigns about the importance of walking dogs, as an example, than they would a novel scientific breakthrough portrayed in the media. Allum, Sturgis, Tabourazi, and Brunton-Smith [108] proposed that the scientific knowledge and people’s attitudes towards science change through the different domains of science and technology. Factors, such as values, emotions, ideology, social identity, and trust in scientific and other institutions, impact on people’s attitudes to science as opposed to their science knowledge [108]. For most dog owners, factors such as emotional attachment, attitudes toward animals, and social norms are likely to impact on their science knowledge and engagement. Likewise, Baram-Tsabari and Osborne [106] suggested that people make meaning from the science they encounter using different narratives based on culturally relevant prior knowledge, which may or may not include science. The relevance of the subject or individuals’ goals to understanding might depend on that reasoning or vice-versa. Therefore, each individual’s learning from welfare science education or interactions will be different based on their cultural prior knowledge as much, if not more so than the science.

Owners are consumers of science and they need to be able to distinguish between high quality, low quality, and pseudoscience [111]. The relationship between the media and science, where most owners are likely to obtain at least some of their science knowledge can be somewhat confusing [112]. Similarly, the interpretation of research findings by the media and other sources often over-simplifies or misrepresents scientific findings [112]. Even potentially trustworthy sources of information may mislead or be misinterpreted by those who do not question them [113]. Individuals need scientific literacy skills; they need to be able to judge whether evidence is consistent with claims. To do this, they need some knowledge of data collection, data interpretation, the role of modelling, the role of uncertainty, and how data are communicated in the public domain [106]. These metacognitive assessment skills are now developed through science education within learning institutions. However, the effectiveness of these strategies will depend on the successes of individual teaching and learning strategies and continue to be somewhat difficult to evaluate. 

Sinatra, et al. [114] explored three factors that challenge the public’s understanding of science, whilst contextualising the importance of the perceived relevance to the individual. The three factors highlighted included epistemic beliefs or beliefs about knowledge or science, motivated reasoning or reasoning with a goal or purpose, and difficulties dealing with conceptual change. Conceptual change challenges individuals. Individuals are all prone to heuristic thinking; making quick judgements on a superficial level, usually reinforcing something that is already thought or believed [115]. Dealing with information that is in conflict to current thoughts or beliefs or experiences should, in theory, lead to deeper, more analytical thought that is required for change [111]. However rather than deal with the belief incongruency and deeper thought required for belief change, we are likely to disregard the conflicting, often uncomfortable information and not change. For example, separation related behaviours could be a significant welfare problem for some dogs living in homes. According the Office for National Statistics, 76% of people in the UK aged between 16–64 are in employment [116]. Many of these are dog owners who leave their dogs for longer periods of time than their dog may be able to cope with. Despite dealing with house soiling or the results of destructive behaviours each time they return home, those owners often fail to recognise this as a result of their dog being left or a consequence of their own behaviours not meeting their dog’s welfare needs. Owners might justify their behaviours as their dog being ‘naughty’ or ‘bored’. Owners frequently underestimate how long they actually leave their dogs or see that they have no choice and rather than deal with the problem, attempt to manage it by confining their dogs to small areas, or leaving them in the garden. Similarly, owners often state that their dogs provide companionship, but do not then consider their dog’s need for company when they cannot provide it. Even if these separation-related behaviours are recognised, and help sought from a behaviourist, then their ‘expert’ observations of the dog’s potential suffering or distress and links to owner behaviour can be challenging for owners to hear, and they may subsequently disagree with the ‘diagnosis’ and fail once again to follow up with a suggested management plan.

Equally, informed decision-making may not be based on an evaluation of what is true, but a second-hand evaluation of who to trust, therefore science education needs to teach people not only how to judge (critical engagement) but who to trust [106]. Cummings [117] explored the impact of trust relating to public health, but these findings could be equally applied to trust of any experts. She highlights the perceived lack of trust and lack of credibility of institutions, including the government, industries, policies, and experts, and points out that surveys show trust in authority to be at an all-time low [117]. Bromme and Goldman [104] suggested that trusting or not trusting claims is a way of deferring the need to judge the claims. Believing or not believing a vet or dog trainer might be easier to decide than trying to establish whether something you potentially know very little about is true or not. 

Many of the problems, debates, and challenges in everyday life for dog owners considering their dog’s needs are not always scientific with ‘black or white’ with ‘yes or no’ answers. There are often socio-scientific components to these issues, such as ethics, economics, culture, and psychology. The unknown element and blurring of these challenges make the dog-owning public decision-making even more challenging. It is also worth questioning whether the public views information about dogs as science. As previously stated, dog ownership is frequently seen as a right and a norm in the UK. If owning a dog is easy, do dog owners need educating about the “science” of dogs or dog welfare? Dog science findings are not likely to be perceived to be new or exciting to the public. Equally, with the addition of emotional attachment to dogs, owners are unlikely to positively respond to belief incongruent information, which suggests that they may not have been doing the best thing for their dog. They are more likely to respond to peer pressure, societal norms, habits, and emotions than scientific facts.

## 10. Is Owner Education the Solution to the Challenge?

It seems fair to suggest that, in the context of dog ownership, educating owners has its limits. Successful education is far more complex than just the provision of information. The effectiveness of education as a strategy depends on multiple factors, including the context, timing and environment, the educator or teacher, the communication, the information, and the individual being ‘educated’. If these series of factors combine in the right way then education could be a very effective and powerful tool for certain individuals in specific circumstances. However, it appears that these series of circumstances are increasingly unlikely to present themselves in the current context that dog ownership occurs. Science communication literature shows that owners’ engagement with welfare science as a concept is unlikely to be predictable and that attempting to increase knowledge and understanding might not be as important as changing individuals’ values, attitudes, beliefs, and developing social norms over time [106]. Equally, it remains unclear which education approaches do or can work as interventions are very rarely implemented in a way that allows for the monitoring and evaluation of their effectiveness.

Hence, even if owners know or understand, or do not know or understand, does it matter if we are doing the right or best thing? The relationship between knowing and doing is another element of the complexity of what makes us individuals. For example, we generally all know that smoking, exercising infrequently, and drinking alcohol are bad for us, yet many of us still pursue these activities. There is also the interesting notion of the difference between public and private behaviours, what we do in our own homes can be very different from what we would do or say in public, so admitting to ourselves what we do or think is also another individual complexity worth noting.

Attempting to educate individuals on healthy behaviours for themselves, their families, and their communities has been the purpose of the National Health Service (NHS) in the UK. The NHS has through 70 years of varying approaches and changes in contexts such as laws, policies, and developments in medical and behavioural knowledge, had to adapt and develop its approach to healthcare delivery. It has begun to move away from health to welfare, and from educational, directive, authoritarian approaches to meaningful engagement with individuals. It now emphasises personal responsibility to bring about improvements in health, changes in policy and legislation, and it has developed varying approaches to different circumstances and individualised patient care. Whilst the NHS’s effectiveness on many fronts remains questionable, significant improvements in many areas of healthcare, such as smoking cessation, have been effectively developed through these people-centred approaches [118]. 

It seems that labelling people as good or bad, or fat or thin, or healthy or unhealthy, is ineffective. Criticism of individuals is thought to be unlikely to change views, but simply lead to justifying themselves or their behaviours and ultimately creating a lack of trust and cooperation [82]. Equally, creating cognitive dissonance or a conflict between an individual’s values, attitudes, and beliefs is also unlikely to result in the desired change. For example, if a vet bluntly tells an owner that their dog is obese and they need to do something about it, this is unlikely to change that owner’s behaviours and ultimately improve the dog’s long-term welfare. However, by taking the approach of respecting individuals’ choices, empathising, and attempting to understanding other’s perspectives and why they do what they do, even if we do not agree with them, we can then build a more constructive effective strategy of communication and cooperation. Working to change habits and behaviours, improving self-efficacy, as well as knowledge and understanding eventually improves the welfare of dogs living in homes.

One such approach that is utilised by the NHS among others has focused on human behaviour and behaviour change. The big breakthrough for behaviour change came in 2011 when the House of Lords Science and Technology Select Committee commissioned an inquiry and report into ‘Behaviour Change’ with a view of understanding and effectively utilising some of the behaviour change theories in future policy development [119]. However, there is a vast number of theories of behaviour change, many of which lack application and testing. Probably the most well-known, rigorously researched and comprehensively tested behaviour change model was developed by Michie, et al. [120]. The Behaviour Change Wheel utilises a behaviour system at its centre, combined with intervention functions and policy categories to enable effective interventions to be developed and evaluated [120]. This model has successfully been applied to the problems of smoking cessation and obesity prevention within the NHS [118]. However, there remains no agreement, even within the NHS or government departments, as to the most effective model or approach to human or population behaviour change to date.

The concept of ‘One Health’; the integration of human and other animal health and medicine, as well as the environment and social contexts in which they exist appears to be starting to break down the somewhat artificial disciplinary boundaries and promoting multidisciplinary thinking and working [121,122]. Even more recently, the concept of ‘One Welfare’ [123] developed the approach of working together to enhance animal welfare and human wellbeing. This ideal emphasises the need for shared research and learning. Hence, in the spirit of the One Health and One Welfare approaches utilising the experience and lessons learnt from the NHS around changing human behaviour to promote health in people, to facilitate changing dog owners’ behaviours for the good of animals seems logical. It is clear that animal welfare science and veterinary medicine, despite continuing to agree that this is an effective strategy, actually need to incorporate people-based research evidence, if they are to continue to improve to welfare of all animals.

## 11. Conclusions and Animal Welfare Implications 

The number, location, and welfare of most dogs living in homes are unknown. It is clear that, while some owners consistently continue to meet all or most of their dog’s welfare needs throughout their dog’s entire life, others may meet all or some of the needs some of the time and others may fail to meet some or all of their dog’s needs for the majority of the time. Whilst nationwide surveys and research studies have identified deficits in the level of five specific elements of dog welfare, there remains limited information regarding how effective most owners are at meeting their individuals’ dog’s welfare needs. This lack of information in itself makes evaluating the success or failure any interventions to improve welfare, education, or otherwise, rather challenging. 

Welfare scientists should consider cross-disciplinary, cross-boundary working and utilise examples of good practice, such as those developed in the NHS, in order to improve the welfare of dogs living in homes. Education therefore needs to be part of a systematic, coordinated, comprehensive, evaluated, and multidisciplinary approach to changing not only people’s knowledge and understanding, but their values, attitudes, and beliefs and ultimately their ownership practices and behaviours towards their dogs.

In its current form, is owner education the solution to the challenge of improving the welfare of companion dogs? Perhaps this is unlikely, given the complexity of the current context of dog ownership. Education could be a hugely valuable part of an approach to improving dog welfare, but it is very apparent that it is not likely to be an effective strategy in isolation. Education as a term suggests an oversimplification to the needs of the challenge of improving welfare, but, as a principle, its motivations are well founded and ultimately the approaches taken will establish whether it can be effective in some situations. However, we need high quality research, evaluation of outcomes, and shared learning from both the successful and less successful approaches to establish the role that education can play in improving the welfare of companion dogs.

## Figures and Tables

**Figure 1 animals-09-00662-f001:**
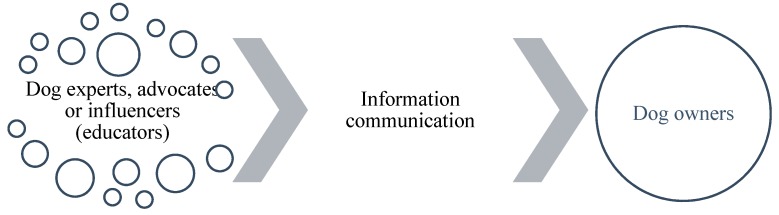
The process of dog owner education.

**Figure 2 animals-09-00662-f002:**
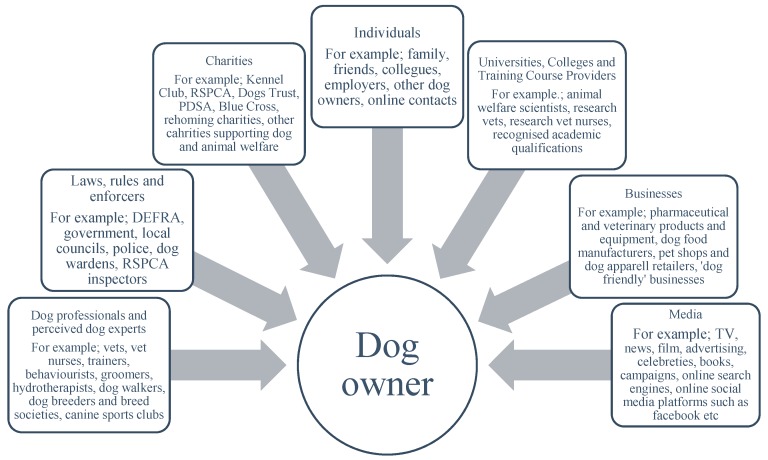
Who, what, or where are the current dog owner educators, teachers, instructors, authorities, communicators, or influencers on dog owners and their dog ownership practices in the UK in 2019?

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
