# Peer review of "Improving the Welfare of Companion Dogs—Is Owner Education the Solution?"

_animals, 2019, doi:10.3390/ani9090662_

Round 1

Reviewer 1 Report

Review: Improving the Welfare of Companion Dogs – is Owner Education the Solution?

I almost never review an article where I don’t think revisions would improve the paper.  However, this is a tight, well-written, and well-reasoned paper and I have no suggestions for improvement.  It is on a very important topic and definitely has made me think about humane education in a different way.  I have recommended humane education for animal welfare issues and now have to think through this too simplistic “solution.”  I thank the authors for addressing this issue.  There is a very small editing issue on page 12 in the paragraph starting at line 520 – there are a few in-text citations rather than footnotes and I don’t think the cites are actually included in the footnotes. 

Author Response

Thank you very much for your review, comments and suggestions. The editing issues highlighted on page 12 line 520 onwards, now line 615 onwards, have been resolved.

Reviewer 2 Report

This paper makes an important contribution to the field, providing a serious and well-documented case for how best to provide education regarding canine behavior. However, it suffer from run on sentences throughout the entire manuscript. I would not say the manuscript needs major rewriting, in that the organization is well done and the presentation is excellent. Instead, I would urge the authors to review their manuscript and break up the longer sentences into two or three shorter sentences throughout the  manuscript. This will make an excellent argument significantly more readable.

Author Response

Thank you very much for your review, comments and suggestions. We have taken them into account and have broken down a large number of sentences and revised our paper accordingly. All changes should be visible as track changes.

Reviewer 3 Report

Brief summary

This review article focused on owner education to improve animal welfare, especially welfare of companion dogs. This study clarified the current problems in the related fields and stated that multidisciplinary approaches are necessary to solve the problems.

Broad comments

This manuscript is interesting and easy to read. The previous studies and information from relevant institutions were cited adequately, which will broaden the scope of readers. A new direction in animal welfare practices can be anticipated in this manuscript.

However, to make the manuscript more useful, please consider the following comments.

It is fine to focus on the UK, but Animals is an international journal. Although the authors have described to some extent in the manuscript, I would like to see the UK's position in the world more clearly. Are the authors’ arguments unique to the UK and applicable only in the UK? Or how far can they be applied in other countries with different social situations, even if it is limited to developed countries? It would be necessary discussion since this study refers to the concepts of social sciences.

For example, do British children learn how to take care of dogs, biology of dogs and animal welfare as moral education in school curricula? What and how do they learn about family maintenance such as childcare and nursing care, which might be an analogy to caring a dog that is considered a family member?

How many British people go out to work, and how much time and money can they spend on caring for their dogs? Does this expenditure balance between welfare of dogs and that of owners?

If people keep dogs because of the therapeutic benefits, what kinds of services are there to support owners technically and financially in the UK as everybody, including socially vulnerable people, can live with their dogs in their home?

What are taught regarding owner education in the formal and regulated training courses to become experts such as veterinarians? Do they learn counseling mind?

It might be rather difficult for foreign readers to understand the British situations consciously without the above-mentioned information. The authors’ arguments will work in a country where the social gap and diversity among the people is small, but there is a question whether the reality is so simple.

The current challenges of animal welfare have been carefully described, but effective solutions to overcome the problems remain as just conceptual framework. Specific solutions should be identified in the future studies and proposing specific solutions might be beyond the objectives of this study. However, readers may feel something lacking or lessen the value of this article as it just states common sense. Is it possible to present one or two concrete suggestions at this stage? For example, it could be easy to understand if there are examples on pages 15-16, lines 715 -719.

Specific comments

Please organize the article list in References according to the Instructions for Authors. In the current version, upper and lower cases of the titles of papers and names of journals are mixed. Official and abbreviated names of journals are also mixed.

38: Date of access is missing.

105: Source of the article?

106: URL?

Author Response

Thank you very much for your review, comments and suggestions. In response to your specific comments:

“It is fine to focus on the UK, but Animals is an international journal. Although the authors have described to some extent in the manuscript, I would like to see the UK's position in the world more clearly. Are the authors’ arguments unique to the UK and applicable only in the UK? Or how far can they be applied in other countries with different social situations, even if it is limited to developed countries? It would be necessary discussion since this study refers to the concepts of social sciences.”

We have attempted to define the UK’s position in the world more clearly in numerous areas. For example; lines 66-69, 141-146, 201-204, 360, 373-421, 427, 433, 775-778. 

“For example, do British children learn how to take care of dogs, biology of dogs and animal welfare as moral education in school curricula? What and how do they learn about family maintenance such as childcare and nursing care, which might be an analogy to caring a dog that is considered a family member?”

Please see lines 141-146.

“How many British people go out to work, and how much time and money can they spend on caring for their dogs? Does this expenditure balance between welfare of dogs and that of owners?”

Please see lines 201-204 and 775-777.

“If people keep dogs because of the therapeutic benefits, what kinds of services are there to support owners technically and financially in the UK as everybody, including socially vulnerable people, can live with their dogs in their home?”

We have not mentioned therapy or assistance dogs in this paper and wonder whether this relationship may be somewhat different. We feel that including something about this may not fit with the contents of the paper and may confuse the argument somewhat.

“What are taught regarding owner education in the formal and regulated training courses to become experts such as veterinarians? Do they learn counseling mind?”

Please see lines 372-421.

“The current challenges of animal welfare have been carefully described, but effective solutions to overcome the problems remain as just conceptual framework. Specific solutions should be identified in the future studies and proposing specific solutions might be beyond the objectives of this study. However, readers may feel something lacking or lessen the value of this article as it just states common sense. Is it possible to present one or two concrete suggestions at this stage? For example, it could be easy to understand if there are examples on pages 15-16, lines 715 -719.”

We consider solutions beyond the scope of this paper, but wonder whether some of the suggestions in section 10 offer ideas and recommendations with examples presented?

“Specific comments

Please organize the article list in References according to the Instructions for Authors. In the current version, upper and lower cases of the titles of papers and names of journals are mixed. Official and abbreviated names of journals are also mixed.”

Referencing style is now consistent according to Instructions for Authors.

“38: Date of access is missing.” date of access is now added.

“105: Source of the article?” source of the article is now added.

“106: URL?” URL now added.

All changes should be visible as track changes.